Whole-genome comparative analysis at the lineage/sublineage level discloses relationships between Mycobacterium tuberculosis genotype and clinical phenotype

Negrete-Paz Andrea Monserrat 1
Vázquez-Marrufo Gerardo 1
Vázquez-Garcidueñas Ma. Soledad soledad.vazquez@umich.mx 2
1 Centro Multidisciplinario de Estudios en Biotecnología, Facultad de Medicina Veterinaria y Zootecnia, Universidad Michoacana de San Nicolás de Hidalgo , Tarímbaro , Michoacán , Mexico
2 División de Estudios de Posgrado, Facultad de Ciencias Médicas y Biológicas “Dr. Ignacio Chávez”, Universidad Michoacana de San Nicolás de Hidalgo , Morelia , Michoacán , Mexico
Flores-Valdez Mario Alberto
Electronic publication date: 2021 Sep 8
Publication date: 2021
Volume: 9
Electronic Location ID: e12128
Received 2021 May 10; Accepted 2021 Aug 17
Copyright: ©2021 Negrete-Paz et al.
Copyright year: 2021
Copyright holder: Negrete-Paz et al.
License: This is an open access article distributed under the terms of the Creative Commons Attribution License, which permits unrestricted use, distribution, reproduction and adaptation in any medium and for any purpose provided that it is properly attributed. For attribution, the original author(s), title, publication source (PeerJ) and either DOI or URL of the article must be cited.
License URL: https://creativecommons.org/licenses/by/4.0/

Keywords: Pulmonary tuberculosis, Extrapulmonary tuberculosis, Drug resistance, Ancient sublineages, Modern sublineages, SNP

Funding: The Coordinación de la Investigación Científica UMSNH Research Program. Graduate scholarship No. 701889 granted by CONACYT-Mexico to A.M.N.P This work was supported by the Coordinación de la Investigación Científica UMSNH Research Program. The funders had no role in study design, data collection and analysis, decision to publish, or preparation of the manuscript.

==============================
Background

Human tuberculosis (TB) caused by members of the Mycobacterium tuberculosis complex (MTBC) is the main cause of death among infectious diseases worldwide. Pulmonary TB (PTB) is the most common clinical phenotype of the disease, but some patients develop an extrapulmonary (EPTB) phenotype in which any organ or tissue can be affected. MTBC species include nine phylogenetic lineages, with some appearing globally and others being geographically restricted. EPTB can or not have pulmonary involvement, challenging its diagnosis when lungs are not implicated, thus causing an inadequate treatment. Finding evidence of a specific M. tuberculosis genetic background associated with EPTB is epidemiologically relevant due to the virulent and multidrug-resistant strains isolated from such cases. Until now, the studies conducted to establish associations between M. tuberculosis lineages and PTB/EPTB phenotypes have shown inconsistent results, which are attributed to the strain predominance from specific M. tuberculosis lineages/sublineages in the samples analyzed and the use of low-resolution phylogenetic tools that have impaired sublineage discrimination abilities. The present work elucidates the relationships between the MTBC strain lineages/sublineages and the clinical phenotypes of the disease as well as the antibiotic resistance of the strains.

Methods

To avoid biases, we retrieved the raw genomic reads (RGRs) of all (n = 245) the M. tuberculosis strains worldwide causing EPTB available in databases and an equally representative sample of the RGRs (n = 245) of PTB strains. A multiple alignment was constructed, and a robust maximum likelihood phylogeny based on single-nucleotide polymorphisms was generated, allowing effective strain lineage/sublineage assignment.

Results

A significant Odds Ratio (OR range: 1.8–8.1) association was found between EPTB and the 1.1.1, 1.2.1, 4.1.2.1 and ancestral Beijing sublineages. Additionally, a significant association between PTB with 4.3.1, 4.3.3, and 4.5 and Asian African 2 and Europe/Russia B0/W148 modern Beijing sublineages was found. We also observed a significant association of Lineage 3 strains with multidrug resistance (OR 3.8; 95% CI [1.1–13.6]), as well as between modern Beijing sublineages and antibiotic resistance (OR 4.3; 3.8–8.6). In this work, it was found that intralineage diversity can drive differences in the immune response that triggers the PTB/EPTB phenotype.

Introduction

Tuberculosis (TB) represents the main cause of death among infectious diseases worldwide, with its drug-resistant manifestations constituting a major global health concern (Floyd et al., 2018). Pulmonary TB (PTB) is the most common clinical disease phenotype, but some patients develop an extrapulmonary TB (EPTB) phenotype in which practically any organ or tissue can be affected, including the aggressive manifestations of lymph node and central nervous system TB (Golden & Vikram, 2005). EPTB represents approximately 15% of reported TB cases globally, whereas as many as 40% of TB cases in several high-income countries are EPTB (WHO, 2020).

Human TB is caused by members of the Mycobacterium tuberculosis complex (MTBC), which have >99% nucleotide sequence identity at the genomic level (Gagneux, 2018). The human-adapted species of the MTBC are M. tuberculosis sensu stricto and Mycobacterium africanum, which are divided into nine phylogenetic lineages: L1, or Indo-Oceanic; L2, or East Asian; L3, or East African-Indian; L4, or Euro-American; L5, or M. africanum West-African 1; L6, or M. africanum West-African 2; L7, or Ethiopia (Firdessa et al., 2013; Coscolla & Gagneux, 2014); L8, or M. tuberculosis from the African Great Lakes (Ngabonziza et al., 2020); and the recently described M. africanum L9 (Coscolla et al., 2021). Global phylogeographic reconstruction of M. tuberculosis suggests that each lineage has become specifically adapted to defined human populations (Gagneux, 2018), with some occurring globally and others showing strong geographical restriction. L4, L2, and L3 are the most commonly found lineages; nevertheless, L4 is the most widespread lineage worldwide. L3 is mostly found in the Middle East, India, and East Africa, while L2 is found predominantly in East Asia (McHenry et al., 2020). This geographical restriction has been found even at the sublineage level, as in sublineage 4.6/Uganda, found only in Uganda and neighboring countries. This sublineage has been shown to possess highly conserved T cell epitopes and a restricted geographic distribution, suggesting a possible adaptation to a specific human population (Stucki et al., 2016). On the other hand, the high incidence of infections associated with nongeographically restricted strains might imply that these strains are more effective in causing the disease (Malik & Godfrey-Faussett, 2005).

Several studies have been conducted to identify possible relationships between M. tuberculosis phylogenetic lineages and the PTB or EPTB phenotype of the disease (Feng et al., 2008; Click et al., 2012), but the results show a lack of consistency. Different factors contribute to explaining the observed discrepancies among the conducted studies in an attempt to establish genotype-phenotype relationships. In the first instance, there were differences in the sample size (Coscolla & Gagneux, 2014), as well as in the nonhomogeneous distribution of lineages, among the set (Kato-Maeda & Nahid, 2012) of analyzed strains. Moreover, biased associations might arise due to a lack of data or failure to control for possible confounders associated with known risk factors for EPTB, such as human immunodeficiency virus (HIV) infection comorbidity in patients from whom strains were isolated (Coscolla & Gagneux, 2014). Furthermore, studies use different operational definitions for EPTB (Kato-Maeda & Nahid, 2012), and some lack appropriate tools to index genomic diversity and classify strains into lineages in some studies (Coscolla & Gagneux, 2010). Whole-genome comparative analysis has allowed the categorization of M. tuberculosis lineages into sublineages using single-nucleotide polymorphism (SNP) analysis (Coll et al., 2014; Stucki et al., 2016). This subtle level of strain differentiation suggests that some sublineages might drive the observed associations of an M. tuberculosis genotype with a specific disease phenotype, such as EPTB (Feng et al., 2008), but an analysis to provide evidence in support of this hypothesis has not been conducted. Interestingly, the frequency of the sublineages assigned to isolated strains from the East Asian (L2) lineage differs among populations settled in different geographical areas, which might also explain why some studies associate L2 lineage strains with EPTB (Feng et al., 2008), whereas others associate it with PTB (Dale et al., 2005), and still others do not find any association at all (Svensson et al., 2011). Thus, intralineage diversity requires a detailed exploration to clarify disease phenotype variation and its relationship with MTBC genotypes (Kato-Maeda & Nahid, 2012). Additionally, genomic evidence has revealed a close relationship among specific M. tuberculosis lineages and sublineages with drug resistance (Wang et al., 2015), a relevant public health phenotype that might hinder successful TB treatment. Horizontal transfer of drug resistance genes has not been reported for M. tuberculosis, but resistance mostly arises from chromosomal mutations under the selective pressure of antibiotic use (Nguyen, 2016). Thus, the characterization of mutations associated with resistance phenotypes in strains with different genotypes can help to reveal lineage-/sublineage-specific microevolutionary processes of epidemiological relevance. Drug-resistance-associated mutations have been hypothesized to modify strain fitness and the ability of a strain to cross the blood–brain barrier, causing EPTB, specifically tuberculous meningitis (Faksri et al., 2018).

Unclarified relationships of lineages/sublineages with M. tuberculosis PTB/EPTB disease and drug resistance phenotypes hinder the generation of adequate epidemiological transmission chains and timely successful treatments. Therefore, this work aims to elucidate relationships between strain genotypes at the lineage/sublineage level and clinical disease phenotypes as well as antibiotic resistance. We retrieved the raw read datasets from the genomes of all the M. tuberculosis strains causing EPTB worldwide and deposited in databases. The raw datasets of the same number of genomes of strains causing PTB isolated from the same countries of origin as EPTB strains were selected to avoid sources of possible biases related to previous studies originating from (i) a low number of analyzed strains, (ii) unequal PTB/EPTB strains analyzed, or (iii) differences in regional clinical TB phenotype incidences. The epidemiological and public health relevance of the identified relationships is discussed.

Materials & Methods

Data retrieval

A total of 490 raw datasets of genome sequence reads were retrieved, which corresponded to 245 M. tuberculosis strains causing PTB and 245 strains causing EPTB (Table S1). We retrieved all available genomes of EPTB strains deposited in the NCBI-SRA database. Such genomes correspond to clinical cases for which the major site of infection reported in databases was not pulmonary or miliary and an additional infection site either was not specified or was specified but not as pulmonary or miliary (Click et al., 2012). The first criterion for the genome selection of the PTB strains was the country of the isolation of the EPTB strains for which the genomes were available. The objective of such a criterion was to analyze the genomes of M. tuberculosis phenotypes from similar clinical and human population backgrounds. The second criterion to select PTB strains was a negative HIV status reported in the associated metadata. In fact, this second criterion was applied for both PTB and EPTB genomes. Additionally, due to the large number of available genomes of PTB strains from different countries, it was possible to discard other comorbidities. All these criteria were used to avoid possible biases generated by population-genotype associations and HIV, or other comorbidities.

The sequence quality of the FASTQ reads was checked using FASTQC (http://www.bioinformatics.babraham.ac.uk/projects/fastqc) and subsequently filtered to a Phred score of 30 using TrimGalore (http://www.bioinformatics.babraham.ac.uk/projects/trim_galore/).

In silico typing

In silico spoligotyping was performed using SpoTyping program version 2.0 (Xia, Teo & Ong, 2016) for next-generation sequencing reads with default parameters. To determine lineage, the TB INSIGHT (http://tbinsight.cs.rpi.edu) server was used based on the identified spoligotypes.

Phylogenetic reconstruction

To find SNPs for phylogenetic reconstruction (Homolka et al., 2012; Coll et al., 2014; Merker et al., 2015), the sequencing reads of the studied strains were aligned to the reference strain of M. tuberculosis H37Rv (accession no. NC_000962.3) using the MTBseq program version 1.0.3 (Kohl et al., 2018) with default values. A frequency of allelic variation greater or equal to 75% and with a phred value >20 was used. Strains where the percentage of reads mapped against the reference genome was less than 80 were excluded due to possible contamination. Also, strains with a median depth coverage >30x were removed. The minimum coverage depth to support a SNP was 8x. For lineage and sublineage assignment, variant positions belonging to repeated regions and resistance genes were excluded. Of the remaining variant positions, those where data quality is below thresholds in >5% of samples were discarded (Jajou et al., 2019). The sublineage phylogeny was rooted using a Mycobacterium microti strain (SRR2667442). Phylogenetic inferences were conducted with the maximum likelihood (ML) criterion using the IQ-TREE package (Nguyen et al., 2015) with a general time-reversible (GTR) model of nucleotide substitution and a gamma model of rate heterogeneity. Phylogenetic trees were constructed based on 1,000 bootstrap replicates, and their visualization was performed using iTOL (Letunic & Bork, 2016).

In silico determination of drug resistance

Raw FASTQ sequencing files were uploaded to TB-Profiler version 3.0.4 (Coll et al., 2015), a tool to determine in silico drug resistance. TB-Profiler can determine genotypic drug resistance by aligning raw sequences against the reference genome M. tuberculosis H37Rv in order to identify mutations (1541 SNPs and indels) previously associated with phenotypic drug-resistance from a curated database. This package also determines the M. tuberculosis lineage based on a 90-SNP barcode. The TB-Profiler-predicted resistance mutations were validated using the results of MTBseq, which reports a list of mutations in genes associated with antimicrobial resistance for every processed strain.

Data analysis

Data entry and statistical analyses were performed in SPSS version 16 (SPSS Inc., Illinois, USA). Univariate analyses were performed using two-tailed Fisher’s exact test to estimate the association between each variable (M. tuberculosis lineage, sublineage or genotypic resistance) and extrapulmonary tuberculosis relative to pulmonary tuberculosis. Additionally, each variable was compared to every anatomical site of TB disease (central nervous system, bones and joints, lymph nodes, and the genitourinary system). Odds ratios with 95% confidence intervals were considered an effect size of the association. To confirm that the association was not an artifact of demographic differences between the geographic regions, a logistic regression model was performed in which the EPTB group was included as a dependent variable. The variables geographic region (the country of isolation), sublineage and genotypic resistance were included in the model, and adjusted ORs with 95% confidence intervals were calculated.

Results

Anatomical site of infection of strains selected for comparative genomics analysis

A total of 490 raw genome datasets of M. tuberculosis strains isolated from seven different countries from individuals with a negative HIV infection status (Table S1) were genotyped. The selected strains included those from countries where specific lineages predominate according to the SITVIT website (http://www.pasteur-guadeloupe.fr:8081/SITVIT_ONLINE/), such as Thailand and Indonesia for the East Asian lineage and India for the East African-Indian lineage. Ninety percent of the selected strains were isolated from Indonesia, Thailand, and Russia. This high percentage is since most of the EPTB strains deposited in the NCBI-SRA database were isolated from these countries. Unfortunately, EPTB strains from African and American countries were not included in the conducted analysis because of the lack of genomes in the NCBI-SRA database until the completion of this study.

The EPTB phenotype strains were clustered into five major groups according to the anatomical site of infection, with a predominance of the central nervous system (72.24%). Other anatomical sites of the disease were bones and joints (17.14%), lymph nodes (4.49%), and the genitourinary system (2.86%); the remaining 3.27% were from sites specified as having only EPTB strains in databases.

Distribution and association of lineages and genotypes with clinical phenotype

The mean coverage obtained for the analyzed strain set was 109.82, whereas the mean percentage of mapped reads was 98.28, resulting in good genome coverage. The strains were assigned to a major M. tuberculosis genetic lineage according to the spoligotype using the TB lineage search option of the TB insight web server. These results correlated with the clustering pattern generated by the SNP-based phylogeny. The phylogenetic analysis of M. tuberculosis strains from PTB and EPTB phenotypes is shown in Fig. 1. The predominant lineage of the analyzed strains was East Asian (54.28%), followed by Euro-American (34.70%), Indo-Oceanic (9.6%), and East African-Indian (1.42%). The EPTB case percentage differed among these phylogenetic lineages, with 15.51% for Indo-Oceanic, 48.98% for East Asian, 2.45% for East African-Indian, and 33.06% for Euro-American. The same variation in phylogenetic lineages was observed for strains from PTB cases, with 59.59%, 36.32%, 3.67%, and 0.40% for the East Asian, Euro-American, Indo-Oceanic, and East African-Indian lineages, respectively. Fisher’s exact test showed that PTB was significantly associated with East Asian lineage strains (OR 1.5; 95% CI [1.1–2.2]) and that EPTB was significantly associated with the Indo-Oceanic lineage strains (OR 4.8; 95% CI [2.2–10.1]) (Table S2). After logistic regression analysis, the associations between L1 and EPTB (OR 4.4; 95% CI [1.9–10.0] P = 0.000) and L2 and PTB (OR 1.4; 95% CI [1.1–1.9]) strains was confirmed, thus eliminating any potential bias due to the isolation regions of M. tuberculosis strains (Table S2).

Figure 1 Phylogenetic analysis of M. tuberculosis strains from PTB and EPTB phenotypes of the disease.

Sensitive, does not present genotypic resistance; Drug-resistant, resistant to at least one antibiotic; MDR, resistant to at least isoniazid and rifampicin. Sensitive, does not present genotypic resistance; Drug-resistant, resistant to at least one antibiotic; MDR, resistant to at least isoniazid and rifampicin; XDR, resistant to isoniazid and rifampicin plus any fluoroquinolone and at least one of three injectable second-line drugs. The phylogenetic tree was inferred using the maximum likelihood (ML) criterion with a general time-reversible model of nucleotide substitution and a gamma model of rate heterogeneity. Yellow highlighted letters indicate EPTB strains. Support values correspond to bootstrap values. The topology was rooted with a Mycobacterium microti strain.

We searched for a relationship between the phylogenetic lineages found in the analyzed strains and anatomical sites of infection, including lungs, lymph nodes, the genitourinary system, the central nervous system, and bones and joints. The results showed that strains belonging to the Indo-Oceanic lineage were significantly associated with central nervous system infection (OR 3.9; 95% CI [2.1−7.4]) (Table S3), whereas those of the East Asian lineage were significantly associated with bone and joint infection (OR 1.5; 95% CI [1.3–1.9]) and with PTB strains, as previously described. Further division of identified lineages was conducted using SNP analysis, distinguishing 27 unique sublineages, with the predominant sublineages being 2.2.1 (n = 218 strains), 2.2.1.1 (n = 25), 4.1.2.1 (n = 18), 4.3.1 (n = 25), 4.8 (n = 51), and 1.2.1 (n = 22). As expected, statistical analysis showed a significant relationship of some of these sublineages with major infection sites. In this way, sublineage 1.1.1 was associated with central nervous system infection (OR 2.8; 95% CI [1.0–7.7]), and the same relationship was observed for sublineages 1.2.1 (OR 6.7; 95% CI [2.0–22.1]) and 4.1.2.1 (OR 6.3; 95% CI [1.9–21.1]). On the other hand, sublineages 4.3.1 (OR 2.9; 95% CI [1.2–7.7]), 4.3.3 (OR 1.8; 95% CI [1.4–2.2]), and 4.5 (OR 3.8; 95% CI [1.0–13.8]) were associated with the pulmonary phenotype.

A deeper whole-genome SNP-based classification to discriminate the proto-Beijing from the Beijing strains allowed us to identify the outbreak sublineages of Asia Ancestral 1, Asia Ancestral 3, Asian African 2, Asian African 2/RD142, Asian African 3, Pacific RD150, Central Asia, and Europe/Russia B0/W148 and a group of modern unclassified strains using informative genetic markers found in the genomes of the studied strains (Fig. 2). Of all these genotypes, the Asian African 2 sublineage (OR 2.3; 95% CI [1.1–5.2]) and Europe/Russia B0/W148 outbreak sublineage (OR 2.7; 95% CI [1.3–5.4]) were significantly associated with PTB. In the same way, the Ancestral Beijing sublineages (Asia Ancestral 1 and Asia Ancestral 3) (OR 2.4; 95% CI [1.2–4.8]) and Central Asia subgroup (OR 8.2; 95% CI [2.9–22.9]) were significantly associated with EPTB.

Figure 2 Phylogeny of 263 Mycobacterium tuberculosis L2 strains.

The phylogeny was constructed by the maximum likelihood (ML) criterion. Classification of strains into sublineages and informative genetic markers are shown. Support values correspond to bootstrap values.

The most frequent spoligotype among L2 strains (85.1%) was SIT1 (Table S4) SIT 53 and SIT 19 were the most frequent spoligotypes for the L4 (22.9%) and L1 (27.6%) strains, respectively. For the L3 strains, 42% had an unclassified SIT spoligotype (703775740003771). Spoligotyping allowed the identification of twelve major genotypic families, including Beijing (53.7%), Central Asia (CAS) (1.4%), EAI (9.8%), Haarlem (4.4%), Latin American–Mediterranean (LAM) (7.5%), S (0.2%), T (19.8%), X (0.4%) and Family33-36 (2%). Interestingly, 13.26% of the strains corresponded to spoligotypes not previously reported in the SITVIT database (Table S4). A significant association was found between the EAI2 family (OR 6.7; 95% CI [2.0–22.1]) and EPTB, of which 13 strains caused tuberculous meningitis and two of them led to lymph node infection.

Mutations associated with drug-resistant TB

A total of 180 previously reported mutations distributed in 18 genes known to confer resistance to first- and second-line drugs for TB treatment were identified (Table S5). Isoniazid was the antibiotic with the highest predicted resistance in 38.77% of the studied strains, followed by rifampicin in 33.46%, streptomycin in 31.02% and ethambutol in 27.14%.

The most common mutation associated with isoniazid resistance was katG Ser315Thr, which was found in 82.63% of the genotypic resistant strains. rpoB Ser450Leu was found in 52.43% of the rifampicin-resistant strains, rpsL Lys43Arg in 72.36% of streptomycin-resistant strains and embB Met306Val in 33.83% of ethambutol-resistant strains.

Extrapulmonary strains showed greater diversity of mutations but appeared less frequently than pulmonary strains. Mutations gyrA (Ala90Val) and thyX (Glu40Gly) were found only in pulmonary strains. On the other hand, mutations in the rpoB (Leu430Pro), rrs (907 A>C), rpsL (Lys88Thr), embA (16C>G), embB (Ala356Val, Ser347Ile), ethA (Gln269*, Gly385Asp), katG (589 insGG), and pncA (Ala3Glu, Cys138Arg, Cys72Arg, Gln10Pro, His51Asp, Gly97Asp, Pro54Leu, Thr135Pro) genes were found only in extrapulmonary strains. There were no statistically significant differences among these variations with the tuberculosis clinical phenotype.

Differences between the distribution of drug-resistant (26.9% PTB, 15.9% EPTB) and drug-sensitive (23.1% PTB, 34.1% EPTB) strains were observed in the EPTB and PTB groups. Four resistance profiles were determined: sensitive, drug resistance (DR), multidrug resistance MDR, and extensive drug resistance (XDR). The frequency for each EPTB/PTB group is shown in Table S2. The bone and joint group presented the highest number of strains with antibiotic resistance: 76.1% were resistant to isoniazid, 73.8% to streptomycin, 66.6% to rifampicin, 54.7% to ethambutol, 42.8% to pyrazinamide and ethionamide, 14.2% to fluoroquinolones and 9.5% to aminoglycosides and paraaminosalicylic acid (Table S6). A total of 13.5% of central nervous system strains showed resistance to isoniazid, and 54.4% of lymph node strains were resistant to rifampicin. Sensitive strains were statistically associated with EPTB (OR: 1.7; 95% CI [1.3–2.3]), and extensive drug resistance (XDR; OR: 6.0; 95% IC 2.5–14.4) was more strongly associated with PTB strains than with EPTB strains. After logistic regression analysis including the geographic region of isolation, XDR strains were still associated with pulmonary rather than extrapulmonary disease (OR: 6.0; 95% IC [2.3–14.4] P = 0.000). We also observed a significant association of Lineage 3 strains with MDR (OR: 3.8; 95% CI [1.1–13.6]) and between the L2 lineage and XDR resistance (OR: 8.1; 5.4−8.6).

Discussion

In this work, we used both raw genome datasets and whole-genome SNP-based phylogeny to genotype the same number of PTB and EPTB M. tuberculosis strains. All the strains were included in one of the six lineages defined by Coll (Coll et al., 2014). When seeking lineage-clinical phenotype relationships, we found that the East Asian ‘modern’ M. tuberculosis lineages were associated with PTB, whereas the Indo-Oceanic so-called an ‘ancient lineage’ were associated with EPTB. It is known that strains from ‘modern’ lineages induced a slighter inflammatory response than those from ‘ancient’ lineages, which has been related to a selective advantage of strains from ‘modern’ lineages, resulting in impaired bacterial control by the host, faster disease progression, and enhanced transmission (Portevin et al., 2011). These clinical differences contribute to explaining the association of the East Asian lineage with the PTB phenotype, supporting the hypothesis that since this is the disease contagious phenotype, patients infected with strains of such a lineage are prone to develop pulmonary disease at a higher frequency than patients infected with strains from other genotypes, which is consistent with the increased transmissibility of strains from this lineage (Thwaites et al., 2018).

It has been proposed that strains belonging to the Indo-Oceanic lineage are ‘less virulent’ than those from other lineages and cause a specific exacerbated inflammatory response (Chakraborty et al., 2018), which might be attributed to the presence of unique cell envelope lipids in this lineage, such as phenol phthiocerol dimycocerosate (Krishnan et al., 2011). Nevertheless, it is still unknown whether these differences in mycobacterial cell envelope lipid composition can explain lineage-related phenotypic differences such as the EPTB phenotype. Interestingly, the percentage of EPTB cases associated with the East African-Indian lineage was as high as 85%, but this finding was not statistically significant due to the small number of EPTB raw genomes (n = 7) from this lineage available in databases. In this regard, the present work reveals database gaps relevant for its public health and epidemiological implications, i.e., the need to include more East African-Indian lineage genomes to clarify its relationships with EPTB.

In the search for a detailed relationship between phylogenetic lineages and the anatomical site of infection, strains belonging to the East Asian lineage were associated with the infection of both bones and joints and PTB. The East Asian lineage comprises two major clades or sublineages, designated proto-Beijing (2.1) and Beijing (2.2) (Ajawatanawong et al., 2019). Sublineage 2.2, or the Beijing family, as defined by spoligotyping, is composed of several sublineages broadly categorized into the ancestral and modern Beijing strains (Mokrousov et al., 2005). A subtle SNP-based classification was recently proposed that allows the discrimination of proto-Beijing from Beijing strains (Shitikov et al., 2017). Such classification divides the Beijing group into the ancestral Beijing clade and the modern Beijing clade, which comprises two groups, one including three strains (Asia Ancestral 1, Asia Ancestral 2, Asia Ancestral 3) and the other including seven strains (Asian African 1, Asian African 2, Asian African 2/RD142, Asian African 3, Pacific RD150, Europe/Russia B0/W148 outbreak and Central Asia). This classification allows us to associate the Asian African 2 sublineage and Europe/Russia B0/W148 outbreak sublineage with PTB, as well as the ancestral Beijing sublineages and Central Asia subgroup with EPTB. Differences in the pathogenicity of Beijing sublineages have been previously reported (Feng et al., 2008), and sublineage specific patterns of induced cytokine production by macrophages have also been observed (Sarkar et al., 2012). In this regard, a macrophage infection model revealed that ancestral Beijing strains induce a higher production of the proinflammatory cytokines TNF-α and IL-6 than the modern Beijing sublineage (Chen et al., 2014). High IFN-γ expression and cytokine production have also been reported in peripheral blood mononuclear cells of ancestral Beijing strains (Faksri et al., 2014). Such results suggest that the ancestral Beijing strains are as highly immunogenic (Kato-Maeda et al., 2012) as Lineage 1 strains (EAI) (Rakotosamimanana et al., 2010). Interestingly, these two sublineages (EAI and ancestral Beijing) were associated with the EPTB phenotype in this study. In contrast, the other sublineage associated with EPTB in this work, 4.1.2.1, has been reported to induce cytokine levels similar to those of H37Rv (Haarlem family strains), which are associated with a low immune response (Wang et al., 2010). This supports the idea that strains of different sublineages vary by many phenotypes, such as the tendency to develop drug resistance, virulence levels, and immune response, which influence disease severity and clinical presentation.

Interestingly, sublineages 4.1.2.1 and ancestral Beijing, belonging to Lineages 4 and 2, respectively, show a high prevalence worldwide, representing more than 50% of the strains in certain areas and/or subpopulations (Ajawatanawong et al., 2019). In contrast, Lineage 1 (EAI) strains are commonly reported in countries around the Indian Ocean and are one of the most geographically restricted families. However, EAI strains have been reported in lower percentages in countries such as the Netherlands, Australia, the USA, Sweden, Saudi Arabia, Tunisia, Taiwan, Panama, and Mexico. The East Asian India 2 spoligotype of Lineage 1 corresponds to the Nonthaburi (EAI2-Nonthaburi) genotype and the Manila (EAI2-Manila) genotype (Couvin, Reynaud & Rastogi, 2019). Recently, Coker (Coker et al., 2016) reported a polymorphism in the genome of Nonthaburi strains from three patients with tuberculous meningitis, consisting of a 500 bp deletion covering ppe50 that was not present in the reference strain M. tuberculosis H37Rv. They reported three mutations (T28910C, C1180580T and, C152178T) until now found only in these meningeal Nonthaburi strains. These mutations could represent part of the genetic background that could be shared by strains that cause EPTB and that also belong to the EAI, 4.1.2.1, and ancestral Beijing sublineages. Nonetheless, further investigation is required to determine whether these mutations are shared, which could be the functional consequences and the probable relation of such polymorphisms with TB disease phenotype.

Regarding the genotype relationship with antibiotic resistance, to the best of our knowledge, a significant association of Lineage 3 with drug resistance found here has not been previously reported. This result might be useful to optimize TB treatment in geographical areas where this lineage is frequent. Studies from Asia, Europe and Africa have shown varying associations between drug resistance and MTB lineages (Coscolla & Gagneux, 2010; Singh et al., 2015); however, as was found here, Beijing strains have been associated with MDR and XDR in several cases (Rodríguez-Castillo et al., 2017). However, it must be highlighted that the lack of experimental drug resistance assays is a limitation of this study. The bioinformatic approach to determine the drug resistance of the studied isolates is based on software packages that use different sets of mutations in their analysis. This generates a variation in the drug-resistant genotypes obtained with the different bioinformatic tools with the consequent risk of over- or underestimating the true drug-resistant behavior of the analyzed strains. The statistical inferences that depend on this determination are also subject to such biases as the association of drug resistance with affected organs or mycobacterial lineages. Thus, it is strongly recommended to perform the experimental determination of the drug resistance phenotype of the strains in which the genome will be deposited in public databases.

The advantages of the present study regarding previous works seeking relationships of the M. tuberculosis genotype with clinical phenotype include the use of a significantly higher number of strains, an equal number of EPTB and PTB strains, a clear assignment of the strains considered to be associated with EPTB, and the exclusion of the strains from HIV patients, with the latter being a controlled variable, a caution not commonly considered in similar works. However, other possible confounders or known risk factors for EPTB could not be considered in the analysis conducted here, partly due to the lack of additional metadata of clinical information in the database where the genomes of the strains were retrieved. This lack of metadata is not a source of bias in the relationships found here because it has been shown that there is an independent association between lineages and EPTB rather than ethnic factors after a stratified analysis (Click et al., 2012). Additionally, it must be noted that the inclusion of genomes previously published that can be generated for different purposes can generate a bias by sampling protocols to seek specific pathogen genotypes or tuberculosis phenotypes. This was a not controlled variable of this study.

These results support the assertion that the relationship between sublineages and clinical disease phenotypes is not attributable to regional-specific factors; therefore, if this association exists, it does not depend only on unknown clinical factors. However, the contribution of the genetic background of the host to the occurrence of the different clinical phenotypes of tuberculosis should not be ruled out. Several polymorphisms associated with EPTB have been previously reported in specific human populations (Fox et al., 2014). Therefore, associations found here must be further tested with additional epidemiological data to clarify the possible relationship between the intrinsic pathogen sublineage characteristics and host factors as genetic background for the establishment of the infection in a particular organ.

Conclusions

Overall, the obtained results suggest that intralineage diversity could drive differences in the immune response that trigger the different clinical phenotypes of tuberculosis. The immune response caused by ancient lineage (L1) and ancestral Beijing strains could be similar, eliciting a high inflammatory response. Our results highlight the need both to analyze the genomic background shared by these strains and to perform in vitro/in vivo studies that help to elucidate the mechanism by which they could disseminate through the body, causing extrapulmonary disease. We demonstrated that the lack of consistency regarding clinical phenotype-strain genotype associations in the results obtained by previous studies is partially due to the use of inadequate phylogenetic classification tools, which do not allow discrimination between sublineages. Additionally, the present results were not biased by the predominance of a specific lineage/sublineage in a specific human population because we included all the EPTB strains available in databases and an evenly representative sample of PTB strains. Biases originating from unknown sampling procedures of the strains for which genomes are available in databases and due to unknown phenotypic drug resistance are the main limitations of the present study.

Supplemental Information

Supplemental Information 1 Raw genome data strains

Click here for additional data file.

Supplemental Information 2 Associations between Mycobacterium tuberculosis lineage, sublineage and genotypic resistance with clinical phenotypes

The frequencies of PTB / EPTB strains by lineage, sublineage and profile of resistance to antibiotics. These data were used for statistical analysis in order to search for possible associations. The results of this analysis are also shown in the table.

Click here for additional data file.

Supplemental Information 3 Associations between Mycobacterium tuberculosis lineage, sublineage and genotypic resistance with anatomic region of isolation

The frequency of the isolated strains of CNS, bone and joints, lymph, genitourinary system, lungs and extrapulmonary tuberculosis without defined anatomical isolation site. The frequencies are described in relation to lineages, sublineages and profiles of resistance to antibiotics and were used for statistical analysis.

Click here for additional data file.

Supplemental Information 4 Identified families, spoligotypes and SITs

The spoligotypes identified in this study, as well as their frequency and the assigned family. These frequencies were used to search for associations between the spoligotype and the clinical phenotype of tuberculosis.

Click here for additional data file.

Supplemental Information 5 Identified mutations in genes that are known to confer resistance to first and second-line drugs

The mutations associated with antibiotic resistance and the frequency observed. These data were used to search for associations between specific mutations and the extrapulmonary phenotype, and to define the assigned resistance profiles.

Click here for additional data file.

Supplemental Information 6 Frequency of extrapulmonary resistant strains by antibiotic

The frequency of strains isolated from CNS, bone and joints, lymph, genitourinary system, lungs and extrapulmonary tuberculosis without defined anatomical isolation site that presented genotypic resistance to first- and second-line antibiotics are shown in Table S6, these data were subjected to a statistical analysis to determine associations between the anatomical site of isolation and resistance to a given antibiotic.

Click here for additional data file.

We appreciate the criticism of Dr. Sebastien Gagneux and an anonymous reviewer that contributed to significantly improve the manuscript.

Additional Information and Declarations

Competing Interests

Author Contributions

Data Availability

The authors declare there are no competing interests.

Andrea Monserrat Negrete-Paz performed the experiments, analyzed the data, prepared figures and/or tables, authored or reviewed drafts of the paper, and approved the final draft.

Gerardo Vázquez-Marrufo and Ma. Soledad Vázquez-Garcidueñas conceived and designed the experiments, analyzed the data, authored or reviewed drafts of the paper, and approved the final draft.

The following information was supplied regarding data availability:

The raw data are available in the Supplemental Files.

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
