# Peer review of "Whole-genome comparative analysis at the lineage/sublineage level discloses relationships between Mycobacterium tuberculosis genotype and clinical phenotype"

_PeerJ, doi:10.7717/peerj.12128_

## Round 0.1 · original submission · Major Revisions

I encourage you to modify your manuscript by taking into consideration both reviewers comments.

·

Basic reporting

In this paper, the authors downloaded publically available whole genome data from Mycobacterium tuberculosis (Mtb) clinical isolates to search for possible associations between Mtb lineage/sublineage and the form of tuberculosis (TB) disease (i.e. extrapulmonary (ETB) versus pulmonary TB (PTB) and antibiotic resistance. They used all available ETB genomes (N=245) and matched these with an equal number of PTB genomes. The find Mtb Lineage 2 (L2) associated with PTB and L1 associated with ETB. Moreover, they find L3 to be associated with multidrug-resistant TB (MDR-TB) and the modern Beijing sublineage of L2 associated with “antibiotic resistance”. My comments are the following:

1) The manuscript would greatly benefit from input by a native English speaker, as there are several instances where the wording is rather awkward.

Experimental design

2) All available ETB genomes were used, but how was the subset of PTB genomes selected?

3) There are insufficient details in the Methods to understand analyses the authors have carried out; this is particularly true for the logistic regression models mentioned. For example, what were the outcome variables; i.e. how where the different lineages and sublineages compared (all against all, or grouped on some way etc.)?

4) Where “clustered” genomes excluded from the analysis? Inclusion of multiple representatives of the same strain might be biasing the analysis.

5) The authors mention a large number of odds ratios in the text but I could not find any table where all these data are properly presented, including the proportion of the different categories that have been analyzed, p-values etc.

Validity of the findings

6) The authors mention correctly in the Introduction that allowing for confounding factors when carrying out these kind of analyses is important. However, it seems that the only patient variable the authors were able to control for was HIV co-infection (i.e. by choosing available genomes from HIV-negative TB patients only). What about variables such as age, sex, other comorbidities, all of which could influence the presentation of TB?

7) Following from the previous comment, one important aspect, which the authors completely ignore, is the potential impact of host genetics. There are indications that populations from South Asia and North Africa (and perhaps elsewhere) might have a higher risk for ETB, completely independently from the infecting Mtb strain. At the same time, and as the authors correctly acknowledge, the Mtb populations are strongly geographically structured. Hence, without considering the host genetic background, the associations with Mtb lineages might be a more consequence of the geographical association between host populations and Mtb lineages. This possibility should at least be discussed.

8) In the abstract the authors state that these association with E/PTB where with “ancient” and “modern” lineages but in the main text they clearly show that this refers to L1 and L2 which is not the same as saying “the ancient and modern lineages”.

Additional comments

9) On multiple occasions, the authors argue that their study is more robust because a lack of potential bias based on their genome selection. This might be partially true. However, by relaying on previously published genomes that were generated for other purposes, new types of (e.g. sampling) bias can emerge, which the authors cannot control for because they are essentially blind to how and why these strains where collected and sequenced in the first place.

Reviewer 2 ·

Basic reporting

Negrete-Paz et al investigated the statistical associations between the types of tuberculosis (pulmonary and extrapulmonary) and lineages / sublineages, in addition to genetic predictors of drug resistance.

Since the study is mainly focused on the analysis of lineages and types of tuberculosis (pulmonary or extrapulmonary), the corresponding rates should be established worldwide or by regions, in such a way that they help to better demonstrate their situation and hence the importance of present study.

Experimental design

The present study does not specify how the selection of strains included in the analyzes was carried out. It only mentions that equal amounts of strains obtained from pulmonary and extrapulmonary tuberculosis were included.

The eligibility criteria that were taken into account should be clarified in the corresponding section. On the other hand, in the methods or discussion section it should be clarified why an adequate proportion of strains obtained in the continents of Africa and America was not included.

It must also be supported why 91% of the strains included in the analysis only come from three countries (Russia, Indonesia and Thailand).

The quality of the reads used in the analysis is not clear. The authors only mention that they were filtered with a Phred score greater than 30. However, in the results section, the authors do not specify the percentages of the reads that were finally filtered or if these reads had good genomic coverage.

All programs must show the version used, since through the years the programs vary due to improvements that may include changes in the internal analysis algorithms. An example of this is that the TBprofiler program used is version 0.2.1, when at present this program is in version 3.0.4. The version used includes a base of 1325 mutations while the last update of this program includes up to a total of 1541. This could affect the prediction of resistance and the consequent association with lineages and sites of infection. Finally, the latest version of the TBprofiler also uses a total of 90 SNPs for the determination of lineages compared to the 62 initially established.

Regarding the obtaining of variants used for the evolutionary reconstruction by the phylogenetic tree and the in-silico determination of drug resistance, the authors only mention the programs used, however it is not specified whether they used the default values of the different options of the same or if these were determined by criteria of the researchers. For example, given that evolutionary inference requires the use of variants that have been fixed in the genome, variants with allelic variants greater than 75% should be considered.

Validity of the findings

In Figures 1 and 2, the name of the figure should be mentioned in their description, as well as that the bootstraps values are established in the respective branches. Also, it is not specified whether any external groups were used to root the tree. In case you have used, it would be appropriate to mention the sequence name (or GenBank accession number). Likewise, in figure 2 it is observed that the ancestral node has up to three Bootstraps values. Is it a mistake or do you have any explanation?

Figure 1 mentions the categories: sensitive, drug-resistant, MDR and XDR. However, the exact definition of them is not clarified in the text. In addition, these 4 categories are considered as "drug resistant phenotype" when in reality the determination of these categories are obtained in silico. In any case, you could be defined as drug-resistant genotypes, since the word phenotype is associated more with a behavior demonstrated in microbiological cultures. In the same figure, it is observed that the labels of some sequences have a yellow background, which is not clarified anywhere in the text.
The statistical values obtained from the main results are mentioned; however, since the statistical association analyzes are a strong component of the study development, the other statistical results obtained from the other variables analyzed should be specified in a supplementary section.

Additional comments

Negrete-Paz et al investigated the statistical associations between the types of tuberculosis (pulmonary and extrapulmonary) and lineages / sublineages, in addition to genetic predictors of drug resistance.

In general, the work is good and detailed, however the recommendations must be taken into consideration in such a way that an improvement is noticed in it.

Finally it is important to mention that a limitation of the study would be the fact that there was no information on phenotypic drug resistance in vitro obtained by any microbial culture method. Instead ofi t, the determination of drug resistance was only inferred by detecting genetic markers in target genes.

Different bioinformatics programs use sets of bases of mutations associated with drug resistance that can vary from one another. This produces a variation in the drug resistant genotypes obtained with the consequent risk of over or under estimating the true drug resistant behavior of the strains analyzed, as well as the statistical inferences that depend on this determination (eg association of drug resistance with affected organs or mycobacterial lineages)

---

## Round 0.2 · Minor Revisions

Please consider these suggestions to improve your work.

·

Basic reporting

The authors have addressed all my comments. I have only 1 small additional comment:

In the Introduction the authors refer to L8 as "M. africanum", which is incorrect. Please remove this statement. Based on the current limited knowledge, we just don't know what the host tropism of L8 is and thus whether to refer to it as "M. tuberculosis sensu stricto" or anything else. It clearly is a member of the Mtb complex though.

Experimental design

ok.

Validity of the findings

ok.

Additional comments

ok

---

## Round 0.3 · accepted · Accept

Congratulations on your achievements. I am not sure whether you can include in the acknowledgement section the names of reviewers. I will make this note to the staff.